# Opportunities for Nanomedicine in *Clostridioides difficile* Infection

**DOI:** 10.3390/antibiotics10080948

**Published:** 2021-08-05

**Authors:** Pei-Wen Wang, Wei-Ting Lee, Ya-Na Wu, Dar-Bin Shieh

**Affiliations:** 1School of Dentistry and Institute of Oral Medicine, National Cheng Kung University, Tainan 701401, Taiwan; 9503049@gs.ncku.edu.tw (P.-W.W.); yana.wu@gmail.com (Y.-N.W.); 2Center of Applied Nanomedicine, National Cheng Kung University, Tainan 701401, Taiwan; 3Department of Obstetrics and Gynecology, National Cheng Kung University Hospital, Tainan 704302, Taiwan; wesker1206@gmail.com; 4iMANI Center of the National Core Facility for Biopharmaceuticals, Ministry of Science and Technology, Taipei 701401, Taiwan; 5Department of Stomatology, National Cheng Kung University Hospital, Tainan 704302, Taiwan; 6Core Facility Center, National Cheng Kung University, Tainan 701401, Taiwan

**Keywords:** *Clostridioides difficile*, spores, anti-spore, spore germination, nanomaterial

## Abstract

*Clostridioides difficile*, a spore-forming bacterium, is a nosocomial infectious pathogen which can be found in animals as well. Although various antibiotics and disinfectants were developed, *C. difficile* infection (CDI) remains a serious health problem. *C. difficile* spores have complex structures and dormant characteristics that contribute to their resistance to harsh environments, successful transmission and recurrence. *C. difficile* spores can germinate quickly after being exposed to bile acid and co-germinant in a suitable environment. The vegetative cells produce endospores, and the mature spores are released from the hosts for dissemination of the pathogen. Therefore, concurrent elimination of *C. difficile* vegetative cells and inhibition of spore germination is essential for effective control of CDI. This review focused on the molecular pathogenesis of CDI and new trends in targeting both spores and vegetative cells of this pathogen, as well as the potential contribution of nanotechnologies for the effective management of CDI.

## 1. Introduction

*Clostridioides* (formerly *Clostridium*) *difficile*, a Gram-positive bacterium that causes severe antibiotic-associated diarrheas and colitis, was first isolated from new-born infants in 1935 [1]. It produces oval terminal endospores [1] and is commonly acquired through community and hospital (nosocomial) infections [2]. The disease is associated with inappropriate antibiotic treatment, which causes an imbalance of the host’s intestine microbial flora, in turn activating the dormant *C. difficile* [3]. Clindamycin, carbapenems and fluoroquinolones are the antibiotics most commonly associated with increasing the risk of *C. difficile* infection (CDI) [4]. Additionally, gastric acid-suppressant and older age (>65 years) are also important risk factors [5]. The symptoms of CDI include watery diarrhea, fever, abdominal pain and toxic megacolon [6]. From a recent epidemiological analysis reported by NHS trusts in England, there were a total of 13,177 CDI cases diagnosed between 2019 and 2020 [7], a small increase of 7.4% compared to the previous year (*n* = 12,274). The incidence of hospital-onset CDI cases mirrors the trends in the incidence of all cases, with a decline between 2007 and 2014 followed by a relatively stable state till 2018. The rate (hospital-onset cases/100,000 bed-days) of hospital-onset CDI cases increased from 12.2 to 13.6 between 2018 and 2020. In the USA, a statistical analysis revealed that CDI cases in 10 hospitals increased between 2011 and 2017, while the adjusted estimate of the burden of hospitalizations for CDI decreased by 24% [8]. Whereas the adjusted estimates of the burden of first recurrences and in-hospital deaths did not change significantly, suggesting the effectiveness of infection-prevention practices, and new more refined diagnostic techniques, it is important to eliminate false positives and to improve infection prevention. However, reducing the burden of CDI remains one of the imperative health care priorities in western countries. Different ribotypes occur according to geographic localization and time of the episodes, associated with evolutionary sophistication. Ribotype 027 and 078 strains have spread worldwide since the millennium, and this was attributed to their ability to metabolize disaccharide trehalose approved by the USA Food and Drug Administration (FDA) since 2000 [9]. Epidemic *C. difficile* ribotypes yield more toxins and have higher sporulation compared to non-epidemic ones [10,11], in spite of some controversial results [11].

The *C. difficile* spores play an important role in its pathogenesis and are well known to resist gastric acid, harsh environments and antibiotics treatment, or even survive in dry inorganic surfaces for months [12]. Hand washing has been recommended as a good practice to reduce risk of pathogen transmission, not only among health care workers but also visitors [13,14]. Hand washing with soap and water is significantly more effective at removing *C. difficile* spores than alcohol-based hand rubs [15,16]. The unique structure of the spores helps the pathogen to overcome UV-A and UV-B irradiation, heat (up to 71 °C), extreme freezing, biocides, chemical disinfectants, desiccation and nutrient deficiency [17,18]. The exosporium is the outermost layer of the spore, containing cysteine (CdeC)-rich proteins that enhance their surface adhesion and spore-host interactions [19,20] and were demonstrated to assist in resistance to heat, enzymes and macrophage-inactivation [19,21]. The next layer is the coat, which blocks oxidizing agents, hypochlorite and enzymes from damaging the microorganism [18,22]. Inside the coat is the outer membrane, and the cortex which keeps the spore in a dehydrated state [23]. The low permeability of the inner membrane prevents the core from invasion from water and other small molecules [24]. The innermost part of the spore is the dehydrated core, containing DNA, RNA, ribosomes, small acid-soluble spore proteins (SASPs) and large amounts of calcium dipicolinic acid (Ca-DPA) [18,25]. The high level of the Ca-DPA increases their resistance to environmental stressors, such as disinfectants and ultraviolet radiation [26].

Several physical methods and chemical reagents have been developed to eliminate spores. For examples, UV-C irradiation (254 nm) could be an ideal option for *C. difficile* spore elimination in a health setting [27]. Moist heat at 85 and 96 °C could effectively kill *C. difficile* spores in foods [28]. Several chemicals have also exhibited anti-spore activity. Sodium hypochlorite (NaOCl) (10%) is a disinfectant with excellent capability to eliminate spores [29]. Hydrogen peroxide (H_2_O_2_) (10%) also has good anti-spore properties [30]. Potassium peroxymonosulfate (KHSO_5_) (0.2%), often used to clean water, is also effective [31]. Nevertheless, these approaches readily used in routine equipment surface and environment disinfection could not be applied for clinical treatment of *C. difficile* spores in human infection. This review provides an update on novel materials that inhibit *C. difficile* and harbor therapeutic potential.

## 2. Molecular Pathogenesis of *C. difficile* Infection

Transmission of *C. difficile* spores or vegetative cells occurs through the fecal—oral route, and from direct contact with contaminated items [32]. However, only spore-form *C. difficile* can pass through the gastric acid and achieve residence in the large intestine [6]. As the spores are exposed to an appropriate environment containing bile acid and co-germinant, they can be reactivated into the vegetative state [33,34]. *C. difficile* spores germinate mostly within the ileum due to the higher environment pH (around 7.4) [34]. At a molecular level, CspC serves as the bile acid germinant receptor, while CspA acts as the co-germinant receptor. Upon activation, the signal is transmitted to CspB [35,36,37], which converts pro-SleC into its active form to degrade the cortex [38,39]. This leads to expansion of the germ cell wall and rehydration of the core, together with the release of dipicolinic acid (DPA) [40,41]. The outgrowth of *C. difficile* spores into a vegetative cell is the result. The vegetative form of *C. difficile* proliferates and produces toxin A (TcdA) and toxin B (TcdB), which contribute to the major pathogenesis process. TcdE protein contributes to the secretion of TcdA and TcdB whcih promotes *C. difficile* growth by obtaining nutrients from toxin-mediated collagen degradation and suppression of competitors in the gut [42,43,44]. Secreted TcdA binds to carbohydrates on the apical surface of colonic epithelial cells, while TcdB recognizes their Wnt receptor frizzled family (FZDs) proteins [45]. Both TcdA and TcdB toxins enter cells via endocytosis, followed by fusion with the lysosome [46,47]. Upon acidification of the organelle, protonation triggers conformation changes in TcdA and TcdB to form hairpin pores; this is followed by release of the glucosyltransferase domains of the toxins from the organelles via autoproteolysis [48,49]. These domains further glycosylate Rho and Rac in the cytosol, preventing them from being activated by guanine nucleotide exchange factors (GEFs), thereby triggering apoptosis and loss of tight junction integrity in mucosal epithelial cells [46,50].

*C. difficile* cells start to sporulate when nutrients are scarce in the environment, although the quorum-sensing signals to relay the environmental state are yet to be identified [25]. Stage 0 sporulation protein A (Spo0A) is a transcription factor critical for the *C. difficile* life cycle that regulates genes associated with biofilm formation, metabolism, toxin production and sporulation [51]. Phosphorylation of Spo0A is an early triggering factor for sporulation, followed by the activation of sigma factor F (σF) to control the downstream effectors σG in the forespore compartment. Activation of σG is required for spore cell wall synthesis and cortex formation [52]. Spo0A also activates σE and further activates σk to modulate coat protein expression and DPA synthesis, as well as their structural assembly after asymmetric division. At this stage, the replicated DNA is already packed into the forespore [53]. After completing the development of the membrane, spore coat and cortex proteins, the mother cell will be lysed, and the mature spore will be released (Figure 1).

Symptomatic *C. difficile* patients shed out the vegetative cells and spores leading to contamination of their environment [54]. The vegetative cells can survive 6 h in room air, while the spores may remain alive for as long as 5 months [55,56]. As the health-care workers and patients’ family members contact the spore-contaminated surface, the spores tightly attach to the skin [57]. Noticeably, asymptomatic *C. difficile* carriers can shed out spores and cause another CDI outbreak [58]. Prevention of *C. difficile* spore formation is, therefore, an important strategy for CDI management and could reduce the threat of relapse [59].

Biofilm formation by vegetative cells and spores has been identified in several Clostridium species, including *Clostridium perfringens*, *C. thermocellum* and *C. acetobutylicum* [60]. Biofilm formation could play major roles in all phases of CDI, especially in their recurrence [61,62], since it helps to enhance microorganism retention, enabling them to resist the flow of luminal material in the gastrointestinal (GI) tract and to prevent the host immune system’s attack. Biofilms provide a powerful shield against antibiotics and create a comfort zone for the microorganism to survive and prosper [63]. Intriguingly, biofilms could also reduce germination efficiency in *C. difficile*. Such controversy complicates the development of treatment strategies.

## 3. Advancements in the Treatment of *C. difficile* Infection

Even though antibiotic therapy is the major treatment of choice for CDI, severe side effects and resistance remain unsolved. Metronidazole and vancomycin have been considered effective treatments for CDI patients for years. However, the two antibiotics also disrupt the normal colonic flora [64] and recurrence is not uncommon [65,66]. Antibiotics also promote spore formation and shedding [67]. Moreover, strains resistant to the two antibiotics were isolated [68] and could spread widely [69]. Fidaxomicin was approved by the FDA in 2011 as a new CDI treatment option, as it showed superior efficacy with significantly less impact on normal colonic flora than vancomycin [70]. Fidaxomicin treatment also inhibits sporulation and decreases spore shedding into the environment [71,72]. Despite only one resistant isolate being reported to date [73], development of non-antibiotic therapy to prevent therapeutic resistance is urgently needed for CDI control.

Some non-antibiotic approaches were developed recently, such as the introduction of probiotics, fecal microbiota transplantation (FMT), engineered microorganisms, bacteriophages, diet control, natural active substances and nanomedicines. A microbiota-based non-antibiotic drug, RBX2660, has been developed by Rebiotix Inc. Clinical trials revealed that, although RBX2660 treatment reduced the number of antibiotic-resistant microorganisms, these still occurred [74]. FMT refers to the transference of fecal materials from healthy donors to patients and was first introduced in 1958 for the treatment of pseudomembranous colitis [75]. Further clinical studies showed that FMT successfully cured patients with recurrent CDI and attenuated CDI-associated diarrhea [76,77]. According to preclinical and clinical data, FMT in combination with vancomycin was recommended as the primary therapy for multiple recurrent CDIs (rCDIs) [78,79]. FMT not only restores healthy gut flora in CDI patients but also interferes with *C. difficile* spore germination [80]. The mechanism by which FMT battles *C. difficile* spore germination is through restoration of secondary bile acid metabolism by bile acid-metabolizing microbiota in the colon and repair of the gut barrier [80,81]. FMT is generally considered a safe treatment modality for rCDIs with only mild side effects (e.g., abdominal discomfort and transient mild fever) [82]. However, before CDI patients receive FMT therapy, both donors and recipient need to undergo rigorous checkups and tests, including evaluating the presence of metabolic syndromes and screening for fecal pathogens [83,84]. In order to improve the clinical application and safety of FMT, further studies should be performed to establish the gold standard of FMT. In addition to FMT, the use of probiotics for CDI prevention and treatment received particular attention in the clinic [85]. Delivery of appropriate probiotics into the intestinal tract could restore the balance of gut microbiota. Recently, mixed regimens containing *Lactobacillus* species, *Saccharomyces boulardii* or *C. butyricum* have been extensively explored for the prophylaxis of CDI [86]. Both probiotics and FMT are high-potential alternative strategies to rebalance the microbiota for effective clinical management of CDIs. However, FMT and probiotics still require an extended treatment course and would not be eligible for all CDI patients, rendering the unmet needs of others to be met by advanced therapeutic options.

## 4. Alternative Strategies for Targeting Spores

The spores of *C. difficile* can be found in food, in domestic animals, and on the surface of contaminated equipment [87] and are difficult to eliminate. Therefore, *C. difficile* endospores are the main vehicle of infection, and anti-spore strategies are undoubtedly important for both disease prevention and therapeutics. These strategies could directly aim at the structural components of the spore or their germination process, as germinating spores are very vulnerable [88]. Many chemicals inhibit *C. difficile* spores.

Ceragenins are synthetic bile acid-based mimics of antimicrobial peptides with broad-spectrum coverage [89,90]. Ceragenin (2 µM) alone significantly reduced biofilm formation by clinical *E. coli* strains, and the combination of ceragenin with LL-37 peptide (10 µM at 1:1 ratio) eliminated 79% of the *E. coli* strains [91]. Remarkably, ceragenin CSA-13 has been shown to have sporicidal activity against *Bacillus subtilis* and *C. difficile* [92,93]. CSA-13 at 75 µg/mL inhibited all *B. subtilis* spore outgrowth by disrupting the inner membrane of *B. subtilis* spores, leading to the release of Ca-DPA from the core followed by premature hydration [92]. CSA-13 affected the germination and viability of *C. difficile* spores even at a lower dose (4 µg/mL), with minimum inhibitory concentration (MIC) of 60 µg/mL [93]. Furthermore, CSA-13 (3 µM) reduced toxin A-mediated inflammation and prevented vancomycin-dependent CDI relapse (10 mg/Kg oral daily) [93]. CSA-13 has an LD_50_ of 24.74 μg/g body weight in mice [94].

Ursodeoxycholic acid (UDCA) is another spore germination inhibitor that acts through a bile acid-based mechanism. When *C. difficile* spores were spread on BHIS agar plates containing 0.1% UDCA (about 2.5 mM), the CFU recovery rate was below 0.0001% [33]. As low as 0.2 mM of the compound is effective to block *C. difficile* spore germination and, at 2 mM concentration, it could even interfere with vegetative cell growth [94]. Weingarden recently reported the successful treatment of CDI patients with UDCA (300 mg twice daily—300 mg four times daily) without relapse [95]. UDCA does not have the 12α-hydroxyl group important for sensing bile acids as a germinating signal by *C. difficile* spores [96]. More bile acid analogues have been chemically synthesized and evaluated for their ability to prevent *C. difficile* spore germination [97]. Some bile acid analogues directly bind to TcdB causing their conformation to change and lose cell binding ability [98]. These results point towards a new pathway for bile acid-based clinical management of CDI patients.

Anti-microbial peptides have also been investigated for inhibiting spore germination. Ramoplanin, a glycolipodepsipeptide antibiotic, was discovered to effectively block bacterial cell wall biosynthesis and presented antibacterial activities against methicillin-resistant *Staphylococcus aureus* and vancomycin-resistant *Enterococcus* [99]. Ramoplanin (50 mg/kg/day) treatment more prominently reduced recovery of *C. difficile* spores in the caecal contents of experimental animals, and cytotoxin production compared to vancomycin at the same dose [100]. *C. difficile* spores exposed to ramoplanin (300 μg/mL) failed to grow out to form colonies on agar plates, through interaction of the compound with the exosporium to ambush their germination [101].

Nylon-3 polymers are another artificial mimic of host-defense peptides synthesized via various β-lactams modified with cationic and lipophilic units [102]. Nylon-3 polymers demonstrated antimicrobial activities against both fungi and bacteria through cell membrane disruption [102,103]. In CDI, nylon-3 polymers not only killed the vegetative cells (MIC: 12.5–25 µg/mL) but also blocked *C. difficile* spore outgrowth (outgrowth inhibitory concentration: 3.13–12.5 µg/mL) [104] rather than directly damage the spores.

An FDA approved bacteriocin called nisin has been used to preserve food for decades [105,106]. Nisin and its analogues have been reported to show anti-bacterial activity against both Gram-positive and Gram-negative bacteria and had additive and synergistic interactions with antibiotics [107,108]. Furthermore, nisin inhibited *C. botulinum* and *B. anthracis* spore outgrowth [109,110]. Nisin (MIC: 0.8–51.2 µg/mL) also inhibited *C. difficile* vegetative cell growth and blocked spore outgrowth (log reduction > 4) at a concentration of 3.2 µg/mL after the germination started [111,112] via binding to the lipid II, thus interfering with cell wall biosynthesis and disrupting the spore membrane [110]. However, Le Lay’s group showed that only a higher concentration of nisin (25.6 µg/mL) directly decreased viability of *C. difficile* spores to 40–50% [111].

Degradation of bacterial cell wall as an anti-CDI strategy has also been extensively explored. Cell wall hydrolase (CWH) encoded by bacteriophages provokes degradation of bacterial cell wall peptidoglycan [113]. Mondal et al. found that a catalytic domain (glucosaminidase and Nlp60 domain) CWH351-656of the hydrolase encoded by *C. difficile* phage phiMMP01 presented lytic activity higher than the full-length CWH. The fragment killed 100% of *C. difficile* vegetative cells and completely inhibited their spores’ outgrowth at a concentration of 200 µg/mL [113]. Although CWH351-656may have therapeutic potential against CDI, further biocompatibility, immunological, and clinical studies are required.

## 5. Emergent Roles for Nanotechnology in Infectious Diseases

Infectious diseases have emerged as a serious global public health concern, underscored by the rapidly increasing number of drug-resistant strains of existing pathogens and the emergence of new pathogens [114]. Multiple challenges must be overcome in the effective management of infectious diseases. These include the lack of safe and effective medications central for disease treatment. The recent development of nanotechnology attracted significant attention due to its potential for transforming both disease diagnostics and therapeutics. Over the past few decades, intensive research in the field enabled the birth of more and more FDA-approved items in chemotherapeutics, anesthetics, imaging contrast agents, nutritional supplements and others [115]. Infectious diseases are also a major focus in nanomedicine.

The use of nanotechnology to defeat multidrug resistance gained significant global attention as new effective antibiotic development is extremely challenging and costly. Modifications in nanoparticles (NPs) could enable multifunctional purposes and bring about advanced applications in medicine. Nanomaterials could be modified with specific targeting moieties such as antibodies or aptamers, to enhance the therapeutic specificity and minimize collateral damage to healthy tissues [116]. Various types of nanomaterials have been shown to deliver drugs with good releasing profile and improve efficacy. These nanomedicines have been extensively explored for applications in the infectious disease area as well [117,118].

Organic nanoparticles (e.g., liposomes, polymeric, micelles and ferritin) have been used to enhance the bioavailability of therapeutic compounds and to increase their delivery and efficacy [119]. They were developed as drug delivery systems, offering a controlled-release profile and targeting the desired tissues or cells. These nanocarrier systems may control drug release by an excipient that enabled slow dissolution of poorly soluble drug crystals, from the core compartment to the interstitial space. Sustained release can also be obtained by encapsulating drugs in nanocarriers capable of loading both hydrophobic and hydrophilic drugs. Organic nanocarrier systems have been evaluated for the treatment of local infections of the female reproductive tract, lungs and skin. Injectable nanocarriers have also been explored for the systemic delivery of drugs [120]. Regarding the types of pathogens targeted, nanoparticles have also been extensively explored for treating fungal [121], bacterial [120] and viral infections, including by *Candida albicans* and severe acute respiratory syndrome 2 (SARS-CoV-2) [122].

Considerable research has focused on polyester-based organic nanosystems that degrade in the presence of physiological esterases (for example, poly (lactide-co-glycolide) (PLGA) and poly (caprolactone)). Modulated by the alteration of the hydrophobicity of the monomer, polymer chain length and particle size, active pharmaceutical ingredients could be released in a predesigned control manner, via bulk degradation of the polymers to enable drug diffusion [123]. Polymers such as poly (anhydrides), poly (orthoesters), poly (cyanoacrylates) and poly (amides) have also been used in the sustained release design [124]. The combination of polymer-based nanoparticles and antibiotics achieved better antibacterial activity than antibiotics alone [125]. Liposomes such as MiKasomes (NeXstar Pharmaceuticals) have also been developed to encapsulate drugs for sustained release in the treatment of bacterial infections including *C. difficile* [126]. These vesicles could also load both hydrophobic and hydrophilic drugs to reduce dosing frequency and ease the dosing regimen.

Inorganic nanoparticles (e.g., metals or metal oxides) have also been investigated for the prevention and treatment of infectious pathogens. Some inorganic nanomaterials were discovered to exhibit diverse activities against multi-drug-resistant pathogens [127]. These include silver (Ag), zinc oxide (ZnO), iron-containing nanoparticles and more. The antibacterial properties of the metallic nanoparticles may be attributed to the generation of reactive oxygen species (ROS), disruption of cell membranes, ability to bind thiol groups (SH-)/disulfide bonds (R-S-S-R) in biomolecules and the release of soluble metal ions [128,129]. The most widely studied metallic materials in infection control are silver (Ag) nanoparticles. The antimicrobial mechanisms of Ag nanoparticles are associated with ROS generation and silver ion release from the nanoparticles. Ag nanoparticles and ions interact with the thiol group, sulfur and phosphorus in the microbial cells subsequently bringing about DNA damage and protein dysfunction [130]. In addition, Ag nanoparticles could also anchor to the bacterial cell wall and cause structural changes in the cell membrane, thus radically affecting cell membrane permeability and inducing cell death. Free radicals generated by Ag nanoparticles upon contact with the bacterial cell membrane are another important mechanism for their anti-microbial activity, as confirmed by electron spin resonance analysis [130]. Compared to their bulk state, Ag nanoparticles also display efficient antimicrobial properties due to their large surface-to-volume ratio, providing better contact interface with the microorganisms [131].

Moreover, nanomaterials responsive to photoactivation have especially been applied for photodynamic, photothermal and photoactivation of chemotherapeutics or their combination. Photoactivation mechanisms could be placed in the payload of the nanocarriers or inside the nanomaterials, as an endogenous property [132]. Photodynamic therapy (PDT) combines special drugs, so-called photosensitizing agents, with light to destroy microorganisms for the management of infectious diseases [133]. The irradiated light activates photosensitizers (PS) to induce the generation of ROS (e.g., peroxides, superoxide, hydroxyl radical, singlet oxygen (^1^O_2_)) that in turn induce cell death [134]. Despite only relatively low laser power being required, successful treatment is dependent upon sufficient oxygen supply to the target tissues. As most PSs are hydrophobic compounds with limited accumulation in the target regions, nanoformulation may provide a solution [135]. On the other hand, photothermal modulation utilizes photo-absorbers to convert photon energy into heat [136]. In this application, high laser power is required and prevention of other side effects such as overheating, which could induce excessive inflammation, should be considered. For photoactivation, photon irradiation was applied to induce a change in the chemical structure, leading to the cleavage of certain functional groups and release of the active pharmaceutical ingredients from the carriers [137]. Over the years, the use of nanomaterials in photothermal therapy has received considerable attention. Inorganic nanoparticles such as zinc oxide (ZnO) and copper oxide nanomaterials have also been reported to harbor strong anti-microbial activities and have been incorporated into a variety of medical applications including skin dressing materials. ZnO has excellent photocatalytic activity [138] and could be accumulated in the microorganisms to cause efficient inhibition of their growth at concentrations between 3 and 10 mM [139]. The lipid bilayer of bacteria is extremely sensitive to ROS. Therefore, photon-induced hydrogen peroxide generation by ZnO plays a primary role in killing bacteria, together with penetration of the cell envelope and disorganization of the cell membrane upon contact with nanoparticles [140].

Nanotechnology has also been applied to disease prevention in vaccine formulation, the so called nanovaccinology. Over the past decade, nanoscale size materials such as virus-like particles (VLPs), liposomes, polymeric, inorganic nanoparticles and emulsions have gained attention as potential delivery vehicles for vaccine antigens, which can both stabilize vaccine antigens and act as adjuvants [141]. These advantages are attributable to the nanoscale particle size, which facilitates uptake by antigen-presenting cells (APCs), leading to efficient antigen recognition and presentation. Modifying the surfaces of nanoparticles with different targeting moieties permits the delivery of antigens to specific receptors on the cell surface, thereby stimulating selective and specific immune responses [142]. The most well-known example is the liposome applied for encapsulating modified mRNAs and stimulating host immune response in SARS-CoV-2 vaccines developed by Moderna and Pfizer-BioNTech [143]. Nanoparticles in the vaccine formulations allow for enhanced immunogenicity and stability of the payload (antigen or nucleic acids encoding the expression of antigen), but also targeted delivery and slow release for longer immunostimulation [144].

Three parenterally delivered vaccines for the prevention of *C. difficile* infection have been developed. These vaccines were based on detoxified or recombinant forms of TcdA and TcdB and are expected to generate high titers of toxin neutralizing antibodies in clinical trials [145]. However, improvements of existing vaccine formulations are necessary. Strategies may include addition of more antigens to limit colonization or sporulation, or integration with treatment regimens. The use of nanotechnology in vaccine development against CDI was first reported in 2017 by Liu’s group using poly-γ-glutamic acid (γ-PGA) and chitosan to form biodegradable nanoparticles [146]. The particle was used to encapsulate recombinant receptor binding domains of TcdB. This strategy successfully induced the production of TcdB neutralizing antibodies in vaccinated mice. The vaccinated mice had low-level inflammation, and all survived the lethal dose of *C. difficile* spore challenge [146]. A nanovaccine for CDI is currently under phase I clinical trial in the USA.

## 6. Nanomaterials for CDI Therapeutics

Organic nanoparticles for the delivery of anti-sense anti-microbial oligonucleotides have been reported recently for anti-CDI therapy [147]. The modified anti-sense oligonucleotides can specifically target five essential *C. difficile* genes simultaneously. They used three (APDE-8, CODE-9, CYDE-21) novel cationic amphiphilic bolaamphiphiles (CABs) to form nano-sized vesicles or vesicle-like aggregates (CABVs) and encapulate 25-mer antisense oligonucleotides (ASO). The empty CABVs had little effect on *C. difficile* growth and could deliver an effective amount of ASO against *C. difficile.* Through encapsulation by bolaamphiphile-based nanocomplex, the oligonucleotides could be effectively transported into *C. difficile* to modulate the translation of specific mRNA which achieved inhibitory concentrations in *C. difficile* without affecting normal microbiota [148,149].

There are reports showing most antibacterial metallic nanomaterials are non-selective generic biocidal agents, mainly against vegetative cells. Their sporicidal activity was explored in only a few studies under high concentrations [150,151]. Ag nanoparticles synthesized by *Streptomyces* sp. have also been reported to exhibit anti-spore potency against *C. difficile* at 75 μg/mL [152]. According to Gopinath’s report, these Ag nanoparticles adhered to the entire spore coat followed by surface protein denaturation and pit formation [152]. Surface modification of Ag nanoparticles with chitosan also showed antibacterial activity against *B. subtilis* vegetative cells and spores [153]. Their activity against *C. difficile* spores remains to be validated. However, the safety for the use of Ag nanoparticles in the human body is still an important concern, as it has been reported that Ag nanoparticles could compromise cell viability and induce pathological damages in animal models [154]. In a clinical study reported by Munger, orally administered Ag nanoparticles (32 ppm) followed by a 2-week observation period presented no clinically important changes in metabolic, hematologic, or urinalysis measures [155]. There were also no detectable morphological changes in the lungs, heart or abdominal organs. ROS formation and subsequent pro-inflammatory cytokine generation commonly associated with Ag nanoparticles were not noted [155]. However, the chronic toxicity of Ag nanoparticles still remains to be further studied [156].

Polyurethane containing crystal violet (CV) and 3–4 nm ZnO nanoparticles have been reported to present bactericidal activity against hospital-acquired pathogens including multidrug-resistant *E. coli*, *Pseudomonas aeruginosa*, methicillin-resistant *S. aureus* (MRSA) and even highly resistant endospores of *C. difficile* [157]. However, recent studies showed that ZnO nanoparticles may affect other microorganisms and impact normal intestinal microflora despite their widespread use in biomedicine [158]. DNA damage induced by ZnO also limits its biomedical application in clinical settings [159]. Other types of nanomaterials responsive to photonic energy (e.g., photodynamic or photothermal therapy) have also been developed for anti-microbial therapy, to enhance drug delivery and local activation. However, the gastrointestinal tract is not the ideal organ for light illumination.

Iron is the most abundant transition metal in the human body and participates in important physiological functions such as oxygen transport and electron transfer. The human body has developed sophisticated systems for the uptake, transport, storage and metabolism of iron. Iron-containing nanoparticles usually exhibited magnetic properties and are among the pioneer nanomaterials used in a wide variety of biomedical and bioengineering applications. Many iron-containing nanoparticles have been evaluated in preclinical and clinical trials, and some of those have reached the market. Zero-valent-iron (ZVI) nanoparticles exhibited excellent biocompatibility while harboring prominently bactericidal efficacy against *E. coli*. Iron oxide nanoparticles also have been widely used for biomedical applications, including hyperthermia therapy and magnetic resonance imagining [160]. There are reports describing that iron oxide nanoparticles have reduced bacterial biofilm formation and viability via an increase in oxidative stress [161,162]. The anti-bacterial mechanisms of these metallic nanoparticles are attributed to their ability to generate reactive oxygen species, disrupt cell membranes, bind thiol groups and release toxic ions. Iron oxide nanoparticles, widely used as T2 weighed MRI imaging contrast agents, were recently used in CDI treatment. The Fe_3-δ_O_4_ magnetite nanoparticles (500 µg/mL) were reported to display sporicidal activity against *C. difficile* spores without adversely affecting the gut microbiota of experimental mice [163]. Fe_3-δ_O_4_ magnetite nanoparticles bind to the surface of *C. difficile* spores, decreasing Ca-DPA release from the spores. The nanoparticles eventually inhibited spore viability in vitro and attenuated *C. difficile*-induced colitis in this mouse model. However, Fe_3-δ_O_4_ nanoparticles did not kill the vegetative cells. A ZVI signal, detected in the Fe_3-δ_O_4_ nanoparticles by X-ray diffraction, has been reported to involve the induction of intracellular oxidative stress and depleting mitochondrial membrane potential in malignant cells [164,165]. Generation of ROS may contribute to the inhibition of *C. difficile* spore germination. As the Fe_3-δ_O_4_ nanoparticles only showed efficacy in anti-spore germination without killing the vegetative form of *C. difficile*, vancomycin-loaded Fe_3-δ_O_4_ nanoparticles (van-IONPs) were synthesized to further enhance CDI control. These nanoparticles demonstrated the ability to inhibit both vegetative cell growth and spore germination [163,166] through direct binding to *C. difficile* spores and blocking their germination, while inhibiting vegetative cells by releasing antibiotics in a synchronized manner (Figure 2). Moreover, van-IONPs protected the intestinal mucosa from *C. difficile* spore adhesion and significantly decreased the level of *C. difficile*-induced inflammation in mice. These nanoparticles outperform both Fe_3-δ_O_4_ nanoparticles and free vancomycin in overall anti-CDI efficacy, due to the dual-function activities of targeting both spores and vegetative cells [166]. ROS production has also been reported as the cause for the anti-bacterial properties of another type of Fe_3-δ_O_4_ nanoparticles [167]. One important advantage of these iron-based nanoparticles in CDI control is their selectivity for *C. difficile* spores and biocompatibility to intestinal mucosa cells. Such properties preserved the normal intestinal flora critical for preventing invasion by other pathogens and protecting the host from recurrence [168].

## 7. Conclusions

*Clostridioides difficile* accounts for about 20% of antibiotic-associated diarrhea, with global incidence increasing significantly between 2001 and 2016. CDI is a common healthcare-associated infection and about 30% of these infections are transmitted within hospitals. Antibiotic therapy is still the major treatment of choice for CDI, despite severe side effects and resistance remaining unsolved. Alternative therapeutic approaches, including FMT, probiotics and other microbiome-based therapeutics delivered through various routes, have been extensively investigated. While novel antimicrobials are being developed against the vegetative cells of the pathogen, non-selective inhibition of bacterial growth may lead to the imbalance of normal intestinal microbiota, leading to recurrent disease and therapeutic resistance. Targeting the spores is a new strategy emerging against CDI, as the *C. difficile* spores are notorious for their resistance to various antibiotic treatments, chemical and physical disinfections, and can survive extremely harsh environment for disease transmission. In this article, we reviewed the efficacy of regimens targeting spore germination, including compounds mimicking bile acids to serve as a decoy to block their germination process, anti-microbial peptides and their bio-mimetic compounds, as well as enzyme fragments to degrade the bacterial cell wall. Nanomaterials provide opportunities to integrate anti-vegetative cell strategies with those targeting spores, while managing toxins produced by these pathogens. While nanomaterials have demonstrated excellent antibacterial capability, they also showed potential adverse effects for gut microbiota, which is important in preventing CDI. Interestingly, iron-containing nanoparticles were recently described by several groups to harbor both antibacterial properties and excellent biocompatibility to intestinal mucosa cells and normal flora. Fe_3-δ_O_4_ nanoparticles have especially showed specific pathogen targeting capability and effective inhibition of spore outgrowth. The next generation of Fe_3-δ_O_4_ nanoparticles carrying vancomycin (van-IONPs) have demonstrated simultaneous targeted inhibition of vegetative cell growth, spore germination and toxin production in a superior fashion to the antibiotics or nanoparticles alone. These advancements prove the great potential of nanomedicine as a novel strategy in the future clinical management of CDI. Tailor-designed nanomedicine will provide new insights and opportunities for precision medicine in the prevention and treatment of emerging infectious diseases.

## Figures and Tables

**Figure 1 antibiotics-10-00948-f001:**
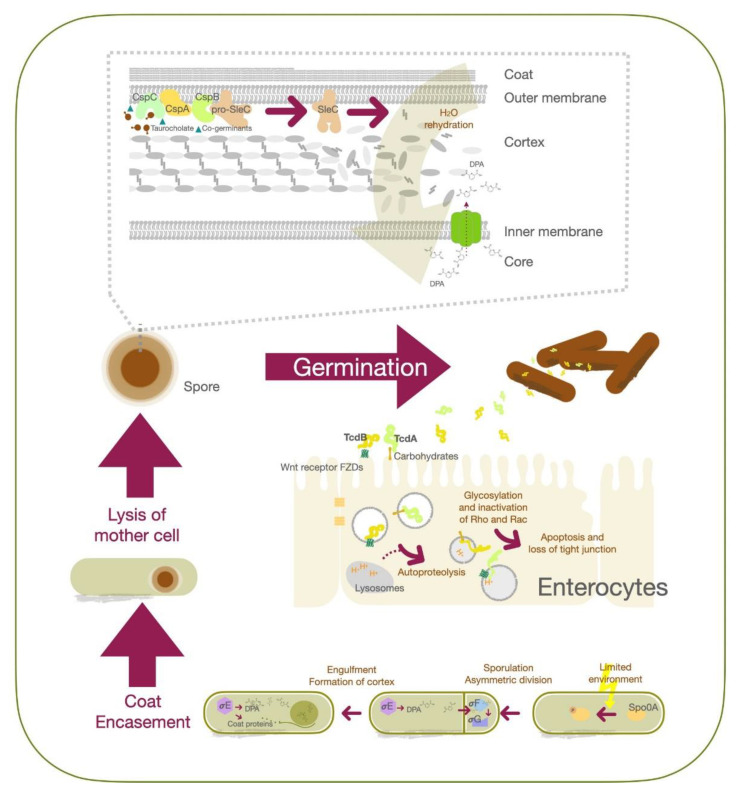
Schematic illustration of *C. difficile* germination, sporulation, and the molecular mechanism of toxins action in epithelial cells.

**Figure 2 antibiotics-10-00948-f002:**
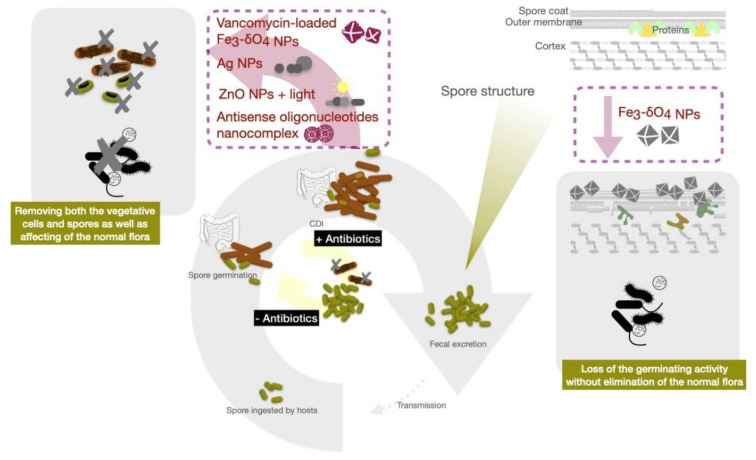
Schematic illustration of nanoparticles (NPs) targeting essential pathways of *C. difficile* infection, including the life cycle of vegetative cells and spores.

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
