# Peer review of "Opportunities for Nanomedicine in Clostridioides difficile Infection"

_antibiotics, 2021, doi:10.3390/antibiotics10080948_

Round 1
Reviewer 1 Report
The review article "Opportunities of Nanomedicine in Clostridium Difficile Infection" by Wang et al. describes modern approaches to treat C. difficile-related infections in humans. The review is relatively well organized and describes major approaches reported in the literature. While the scientific concept is certainly worthy of publishing, the outcome leaves much to be desired.
- Bacterial species names should be in italics. Unfortunately, I have not counted one instance where the name was italicized.
- Many places contain grammatical errors, most likely related to English being the second language. A review by a professional editor would help improve the message and eliminate the errors.
- Nanomedicine-based approaches are only briefly mentioned, mostly on the same level as other approaches. If the manuscript is to be focused on nanomaterial-based approaches, the review section describing such approaches should be extended.
- The effectiveness of approaches in the review was not mentioned. Statements that it was used without giving effective concentrations or percentage of the pathogens killed are, frankly, suitable for something else.
- Nanomedicine as a general approach includes also preventive measures. The review does not mention anything about vaccines based on nanoparticles containing clostridial toxins despite subunits being used routinely in commercial vaccines.
In summary, the review falls short of giving a good background on the biology of C. difficile and the scientific basis for targeting specific components. Mentioning 2 toxins is not sufficient to correct the problem.
Author Response
Response to reviewer 1
Comments and Suggestions for Authors
The review article "Opportunities of Nanomedicine in Clostridium Difficile Infection" by Wang et al. describes modern approaches to treat C. difficile-related infections in humans. The review is relatively well organized and describes major approaches reported in the literature. While the scientific concept is certainly worthy of publishing, the outcome leaves much to be desired.
Bacterial species names should be in italics. Unfortunately, I have not counted one instance where the name was italicized.
Response: Thanks for your valuable suggestions. We have already carefully revised the bacterial species names in italics.
Many places contain grammatical errors, most likely related to English being the second language. A review by a professional editor would help improve the message and eliminate the errors.
Response: We thank the reviewer for the helpful suggestions. We have revised the manuscript and submitted it to a professional language editing service to improve the quality.
Nanomedicine-based approaches are only briefly mentioned, mostly on the same level as other approaches. If the manuscript is to be focused on nanomaterial-based approaches, the review section describing such approaches should be extended.
Response: Thanks for the valuable suggestions. We have reorganized the manuscript to extend the content of the application of nanotechnology in infectious disease management in Section 5. Section 6 was also extended to cover the development of nanotechnology in CDI treatment for the targeting of spores.
The effectiveness of approaches in the review was not mentioned. Statements that it was used without giving effective concentrations or percentage of the pathogens killed are, frankly, suitable for something else.
Response: We have added the effectiveness-related statements, such as concentrations used in various approaches for CDI management.
Nanomedicine as a general approach includes also preventive measures. The review does not mention anything about vaccines based on nanoparticles containing clostridial toxins despite subunits being used routinely in commercial vaccines.
Response: Thank you for the comments. We added a paragraph to introduce the applications of nanotechnologies in vaccine development in lines 353 to 380 of page 9.
In summary, the review falls short of giving a good background on the biology of C. difficile and the scientific basis for targeting specific components. Mentioning 2 toxins is not sufficient to correct the problem.
Response: The pathogenicity of C. difficile is mainly mediated by toxin A (TcdA) and toxin B (TcdB), while genes associated with spore formation and germination also play certain roles. In this manuscript, we present a though discussion of the molecular pathogenesis of both the above-mentioned toxins in lines 104 to 114 of page 3. In addition, it was suggested by other reviewers that this review paper should focus more on nanomaterials in CDI and should only briefly address the biological background of C. difficile.

Reviewer 2 Report
The topic of this study is timely.
The elimination of C. difficile vegetative cells and inhibition of the spore germination at the same time is essential for effective control of CDI. This review focused on the molecular pathogenesis of CDI, new trends targeting spores of the pathogen and the potential contribution of nano- technologies in effective management of CDI.
The article is interesting and contains updated data, but the authors must decide if they intend to carry out a review of all new trends targeting on spore of the pathogen or if they intend to focus their attention on the opportunities of Nanomedicine in Cl. Difficile infection. In the first case, the title of the article should be changed, in the second case, on the newer topic of nanomedicine in Cl.difficile infection, the treatment of paragraphs 2 and 3 and the relative bibliography should be limited.
The manuscript is not suitable for publication in its present form. However, due to the interesting topic, I encourage the Authors to substantial rewrite the article and resubmit it.
Introduction
line 40-41. “The incidence of CDI in the United Kingdom and United States have declined since 2011”
This sentence must be supported by the bibliographic entries and should be better explained or eliminated. I would not report the sentence because it could lead to erroneous consequences. In this form the sentence can be misinterpreted and suggest that the reduction of the observed cases goes in the present day and the problem would be resolved.
If we refer to the references 7 and 8, these cannot be considered suitable for this sentence, but they would fit better in the following sentence, where correctly indicated.
From the quoted NHS publication (reference 7) of Epidemiological analysis of Clostridioides difficile infection the following sentence is extracted: “A total of 13,177 cases of Clostridioides difficile infection were reported by NHS trusts in England between 1 April 2019 and 31 March 2020. This is a small increase of 7.4% from 2018/19 (n = 12,274), and a decrease of 76.3% from 2007/08 (n = 55,498)”.
“The incidence rate for hospital onset CDI cases mirrors the trends in incidence for all cases, with declining rates from 2007/08 to 2013/14 which then remained relatively stable to 2017/18. The rate of hospital-onset CDI cases increased from 12.2 in 2018/19 to 13.6 in 2019/20, a change of 11.7%”.
Also in reference 8 with regard to USA data, it reports that: “The number of cases of C. difficile infection in the 10 U.S. sites was 15,461 in 2011 (10,177 health care–associated and 5284 community-associated cases) and 15,512 in 2017 (7973 health care–associated and 7539 community-associated cases).” In addition, continuing we read “ The adjusted estimate of the burden of hospitalizations for C. difficile infection decreased by 24% (95% CI, 0 to 48), whereas the adjusted estimates of the burden of first recurrences and in-hospital deaths did not change significantly.”
Adherence to recommended infection-prevention practices may have also decreased health care–associated infections. New, more refined diagnostic techniques have made it possible to eliminate false positives, but continued efforts are needed to improve infection prevention as antibiotic stewardship in both inpatient and out patient settings.
line 52-53
“Hand washing has been recognized as a good practice to reduce risk of pathogens transmission”.
I suggest to add the sentence…“not only among health care workers but also among visitors”.
The following bibliographical entries document the statement (if you like):
World Health Organization & WHO Patient Safety (2009). Hand hygiene technical reference manual: to be used by health-care workers, trainers and observers of hand hygiene practices. Available at: https://apps.who.int/iris/handle/10665/44196
Ragusa R, Giorgianni G, Lupo L, Sciacca A, Rametta S, La Verde M, Mulè S, Marranzano M. Healthcare-associated Clostridium difficile infection: role of correct hand hygiene in cross-infection control. J Prev Med Hyg. 2018 Jun 1;59(2): E145-E152.
Scaria E, Barker AK, Alagoz O, Safdar N. Association of Visitor Contact Precautions With Estimated Hospital-Onset Clostridioides difficile Infection Rates in Acute Care Hospitals. JAMA Netw Open. 2021 Feb 1;4(2):e210361.
line 53-54
I would suggest a positive view of the problem such as: Handwashing with soap and water is significantly more effective at removing C. difficile spores from the hands than alcohol-based hand rubs.
It is true that "hand washing with soap alone failed to remove C. difficile spores completely" and residual spores are readily transferred by a hand shake after use of alcohol-based hand rubs but this is the simplest form of contagion prevention we have, given that alcohol is not effective against C. difficile spores.
PARAGRAPH 2.
The role in the pathogenesis of C. difficile spores is well explained.
The paragraph is excessively detailed at some points (line 88-120) and therefore this part could be summarized by the authors.
It is suggested to use acronyms correctly by entering the full name the first time it is used. Furthermore, given the large amount of acronyms, the use of a legend could be useful.
PARAGRAPH 3.
The part describing the new antibiotic treatments and their mode of action is useful and clear. In the part concerning the fecal microbiota transplantation, the description of recent animal study (line 161-165) is not necessary as well as the description of undesiderable events (line 174-176).
PARAGRAPHs 4, 5.
These paragraphs deal with the heart of the problem comprehensively. Review of acronyms is also recommended here.
Figure 1 cannot be presented in the conclusions but should be linked with what is described in the text paragraph 5.
References
The reference list covers the relevant literature adequately and in an unbiased manner. Authors should double check if all references are indicated according to the editorial staff's instructions.
Finally there are some simple typos that can be corrected (es. line 148 - in2011).
Author Response
Response to Reviewer 2
Comments and Suggestions for Authors
The topic of this study is timely.
The elimination of C. difficile vegetative cells and inhibition of the spore germination at the same time is essential for effective control of CDI. This review focused on the molecular pathogenesis of CDI, new trends targeting spores of the pathogen and the potential contribution of nano- technologies in effective management of CDI.
The article is interesting and contains updated data, but the authors must decide if they intend to carry out a review of all new trends targeting on spore of the pathogen or if they intend to focus their attention on the opportunities of Nanomedicine in Cl. Difficile infection. In the first case, the title of the article should be changed, in the second case, on the newer topic of nanomedicine in C. difficile infection, the treatment of paragraphs 2 and 3 and the relative bibliography should be limited.
The manuscript is not suitable for publication in its present form. However, due to the interesting topic, I encourage the Authors to substantial rewrite the article and resubmit it.
Response: This review intended to focus on the applications of nanotechnology in CDI. We therefore reorganized and extended the content to include more coverage in the nanotechnology era, from general infectious disease management to vaccine development, in Section 5 and Section 6. The latter also covered nanotechnology for CDI treatment in addition to targeting spores.
Introduction
lines 40-41. “The incidence of CDI in the United Kingdom and United States have declined since 2011”
This sentence must be supported by the bibliographic entries and should be better explained or eliminated. I would not report the sentence because it could lead to erroneous consequences. In this form the sentence can be misinterpreted and suggest that the reduction of the observed cases goes in the present day and the problem would be resolved.
If we refer to the references 7 and 8, these cannot be considered suitable for this sentence, but they would fit better in the following sentence, where correctly indicated.
From the quoted NHS publication (reference 7) of Epidemiological analysis of Clostridioides difficile infection the following sentence is extracted: “A total of 13,177 cases of Clostridioides difficile infection were reported by NHS trusts in England between 1 April 2019 and 31 March 2020. This is a small increase of 7.4% from 2018/19 (n = 12,274), and a decrease of 76.3% from 2007/08 (n = 55,498)”.
“The incidence rate for hospital onset CDI cases mirrors the trends in incidence for all cases, with declining rates from 2007/08 to 2013/14 which then remained relatively stable to 2017/18. The rate of hospital-onset CDI cases increased from 12.2 in 2018/19 to 13.6 in 2019/20, a change of 11.7%”.
Also in reference 8 with regard to USA data, it reports that: “The number of cases of C. difficile infection in the 10 U.S. sites was 15,461 in 2011 (10,177 health care–associated and 5284 community-associated cases) and 15,512 in 2017 (7973 health care–associated and 7539 community-associated cases).” In addition, continuing we read “ The adjusted estimate of the burden of hospitalizations for C. difficile infection decreased by 24% (95% CI, 0 to 48), whereas the adjusted estimates of the burden of first recurrences and in-hospital deaths did not change significantly.”
Adherence to recommended infection-prevention practices may have also decreased health care–associated infections. New, more refined diagnostic techniques have made it possible to eliminate false positives, but continued efforts are needed to improve infection prevention as antibiotic stewardship in both inpatient and outpatient settings.
Response: Thank you for the recommendation. We have significantly revised the content according to your suggestions in lines 42 - 55 of page 2.
lines 52-53
“Hand washing has been recognized as a good practice to reduce risk of pathogens transmission”.
I suggest to add the sentence…“not only among health care workers but also among visitors”.
The following bibliographical entries document the statement (if you like):
World Health Organization & WHO Patient Safety (2009). Hand hygiene technical reference manual: to be used by health-care workers, trainers and observers of hand hygiene practices. Available at: https://apps.who.int/iris/handle/10665/44196
Ragusa R, Giorgianni G, Lupo L, Sciacca A, Rametta S, La Verde M, Mulè S, Marranzano M. Healthcare-associated Clostridium difficile infection: role of correct hand hygiene in cross-infection control. J Prev Med Hyg. 2018 Jun 1;59(2): E145-E152.
Scaria E, Barker AK, Alagoz O, Safdar N. Association of Visitor Contact Precautions With Estimated Hospital-Onset Clostridioides difficile Infection Rates in Acute Care Hospitals. JAMA Netw Open. 2021 Feb 1;4(2):e210361.
Response: The sentences were revised according to your precious suggestions in lines 64 - 66 of page 2.
lines 53-54
I would suggest a positive view of the problem such as: Handwashing with soap and water is significantly more effective at removing C. difficile spores from the hands than alcohol-based hand rubs.
It is true that "hand washing with soap alone failed to remove C. difficile spores completely" and residual spores are readily transferred by a hand shake after use of alcohol-based hand rubs but this is the simplest form of contagion prevention we have, given that alcohol is not effective against C. difficile spores.
Response: The manuscript was revised in a positive view accordingly in 66 - 67 of page 2.
PARAGRAPH 2.
The role in the pathogenesis of C. difficile spores is well explained.
The paragraph is excessively detailed at some points (lines 88-120) and therefore this part could be summarized by the authors.
It is suggested to use acronyms correctly by entering the full name the first time it is used. Furthermore, given the large amount of acronyms, the use of a legend could be useful.
Response: Thanks for the suggestions; we have condensed the content in lines 100 - 127 of page 3. A legend also added for figure 1.
PARAGRAPH 3.
The part describing the new antibiotic treatments and their mode of action is useful and clear. In the part concerning the fecal microbiota transplantation, the description of recent animal study (lines 161-165) is not necessary as well as the description of undesirable events (lines 174-176).
Response: The unnecessary descriptions were deleted in lines 168 and 186 page 5.
PARAGRAPHS 4, 5.
These paragraphs deal with the heart of the problem comprehensively. Review of acronyms is also recommended here.
Figure 1 cannot be presented in the conclusions but should be linked with what is described in the text paragraph 5.
Response: We have moved this figure to page 11 line 452, in Section 6.
References
The reference list covers the relevant literature adequately and in an unbiased manner. Authors should double check if all references are indicated according to the editorial staff's instructions.
Finally, there are some simple typos that can be corrected (es. line 148 – in 2011).
Response: Thanks for your advice. We have submitted the manuscript for professional English editing and checked the acronyms and the reference accordingly.

Round 2
Reviewer 1 Report
The manuscript has been improved and the requested section on nanomedicines has been expanded. There are some typographical errors found but they could be easily corrected during the proofreading stage.
Reviewer 2 Report
I appreciate that I had a chance to review this manuscript.
The changes made after the reviewers' suggestions made the article more correct and very interesting.
Paragraph 5 enriches the work substantially.
The authors have adequately expanded the bibliography.